# LaFTer: Label-Free Tuning of Zero-shot Classifier using Language and Unlabeled Image Collections

**M. Jehanzeb Mirza**[†1,2]  **Leonid Karlinsky**[3]  **Wei Lin**[1]  **Mateusz Kozinski**[1]
**Horst Possegger**[1]  **Rogerio Feris**[3]  **Horst Bischof**[1,2]

[1]Institute of Computer Graphics and Vision, TU Graz, Austria.
[2]Christian Doppler Laboratory for Embedded Machine Learning.
[3]MIT-IBM Watson AI Lab, USA.

Project Page: https://jmiemirza.github.io/LaFTer/

## Abstract

Recently, large-scale pre-trained Vision and Language (VL) models have set a new state-of-the-art (SOTA) in zero-shot visual classification enabling open-vocabulary recognition of potentially unlimited set of categories defined as simple language prompts. However, despite these great advances, the performance of these zero-shot classifiers still falls short of the results of dedicated (closed category set) classifiers trained with supervised fine-tuning. In this paper we show, for the first time, how to reduce this gap without any labels and without any paired VL data, using an unlabeled image collection and a set of texts auto-generated using a Large Language Model (LLM) describing the categories of interest and effectively substituting labeled visual instances of those categories. Using our label-free approach, we are able to attain significant performance improvements over the zero-shot performance of the base VL model and other contemporary methods and baselines on a wide variety of datasets, demonstrating absolute improvement of up to $11.7\%$ ($3.8\%$ on average) in the label-free setting. Moreover, despite our approach being label-free, we observe $1.3\%$ average gains over leading few-shot prompting baselines that do use 5-shot supervision.

## 1 Introduction

Vision and Language (VL) models [1–8] recently became the de-facto standard for generalized zero-shot learning enabling recognition of arbitrary (open) set of categories provided with just their text descriptions and without requiring any additional data or training. However, this incredible flexibility comes at a performance cost and all the VL models, including the most widely used CLIP [1], still require additional supervised training (e.g. tuning its vision encoder) to compete with the closed set (of target categories) supervised training methods. Naturally, this incurs undesirable adaptation costs for those, otherwise very versatile and flexible, foundation models. Image annotations are often expensive to collect, especially for legacy vision systems, such as traffic surveillance, quality control, or security.

In this paper we propose LaFTer, an approach for completely label-free parameter efficient finetuning of VL models to a set of target classes. Our goal is to substitute the need for expensive-to-obtain image annotations with unsupervised finetuning of VL models. We show that since the VL models share a common text-image embedding space (due to their contrastive learning objective), there is a possibility to train using embeddings of samples from one modality (e.g., auto-labeled text) and then successfully apply what we trained to classify embeddings of samples from the other modality

---

[†]Correspondence: `muhammad.mirza@icg.tugraz.at`

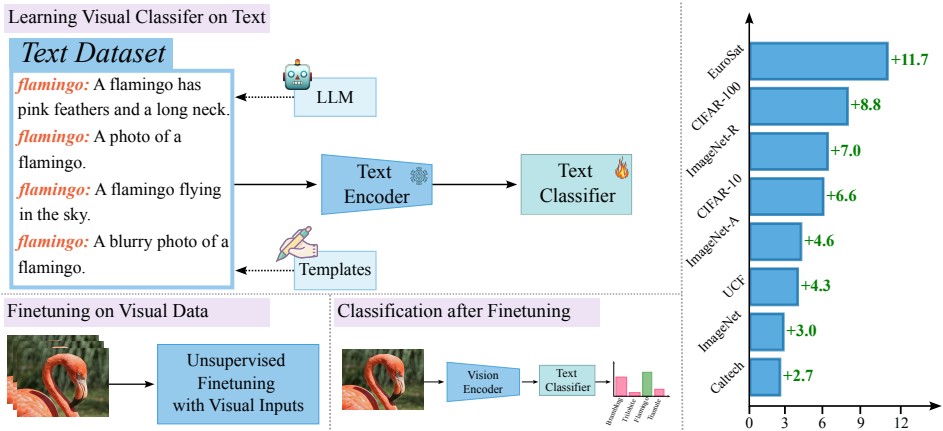

Figure 1: LaFTer proposes to first train a classifier on a natural language text dataset mined in a controlled manner from a set of target classes by generating descriptions for each class label using an LLM and mixing them with handcrafted templates. The training objective is to classify each description to the correct (source) class name (top-left). In the second stage, LaFTer employs the text-only classifier to generate pseudo-labels on the unlabeled data to further finetune the vision encoder in a parameter-efficient manner (bottom-left). Finally, the finetuned visual encoder and text classifier is used for eventual classification (bottom-middle). Combining our text-only pre-trained classifier together with the proposed pseudo-labeling pipeline lets us consistently improve the previous SOTA results for label-free finetuning, UPL [12] (right).

(e.g., unlabeled images). More specifically, we show that it is possible to train a neural network (e.g., a classifier) to classify text instances that can be successfully used to classify images, showing successful cross-modal transfer. Instead of collecting labeled visual instances, in our label-free LaFTer approach, we are mining text descriptions of the target categories by prompting an LLM (e.g., GPT-3 [9]) and combining them with handcrafted prompts, as shown in Figure 1. After creating such a text dataset, we train a neural network to classify each text instance (sentence) in it to the source class label that was used to produce the instance. This text classifier can be readily used to classify images when used on top of a CLIP visual encoder. Furthermore, we take advantage of this text-only pre-trained classifier by employing it in a pseudo-labeling pipeline (inspired by FixMatch [10]), to further finetune the CLIP vision encoder on an unlabeled image collection. To reduce overfitting and keep the finetuning parameter efficient, we make use of Visual Prompt Tuning [11] combined with adapting the affine transformations (scale and shift) of the normalization layers in the otherwise frozen network.

To summarize, our contributions are as follows:

- We offer a new, completely label-free, and parameter-efficient ($0.4\%$ learned parameters), cross-modal transfer approach (LaFTer) to finetuning a VL model for improved recognition performance on a set of target classes using a set of automatically generated texts for those classes and an unlabeled image collection.

- We show that text-side-only training on an automatically collected language knowledge (e.g. through LLM prompting) can be effective in bootstrapping a visual classifier built on top of an aligned VL model to consistently improve its performance in a variety of downstream classification benchmarks.

- In an extensive evaluation of our approach, spanning over 12 popular benchmarks, we show that it: (i) attains up to $28\%$ absolute improvement over the base VL model (CLIP) performance (6.7% on average); (ii) improves over previous SOTA under the same label-free setting by up to $11.7\%$ (3.8% average); and (iii) performs favorably while comparing with SOTA parameter efficient few-shot methods, matching from above average performance of 5-shot supervised setting despite using no labeled samples.

## 2   Related work

**Large-scale Vision&Language (VL) Models:** Remarkable performance in many zero-shot downstream tasks has been attained with VL models pre-trained using contrastive losses on large-scale noisy image-text data (e.g., CLIP [1] and ALIGN [13]). Different ideas have been tried to improve the image-text representation alignment, such as leveraging off-the-shelf object detectors [14–16], or using cross-attention and additional objective functions such as image-text matching and masked language modeling [4, 17–19], or filtering noisy captions (e.g., BLIP [4]). Some additional properties of language structure are utilized in [2, 20–23]. DeCLIP [2] finds additional positives for the contrastive loss by seeking textual nearest-neighbors. Geometrically consistent representations (within and across the paired image-text modalities) are employed in CyClip [20]. Recently, few methods have attempted improving VL performance using additional supervision [3, 24], finer-grained interactions [25], modern Hopfield networks [26], optimal transport distillation [27], cycle consistency [28], and hierarchical feature alignment [29]. However, VL models performance still falls short of classifiers trained with supervision on the downstream target data. Most of the approaches to fine-tuning VL models [30, 31] leverage annotated data for this finetuning. In contrast, in our work we propose a completely label-free approach for adapting a VL model to the target task. Methods which are most related to our work, UPL [12] and CLIP-PR [32] also finetune VL models in an unsupervised manner. UPL finetunes learnable text prompts (similar to [30]) by relying on confidence sampled pseudo-labeling, whereas CLIP-PR relies both on offline pseudo-labeling and label distribution prior from the training data. In contrast, in our work, we first leverage textual knowledge from LLMs and design a self-supervised text-only pre-training task showing that it can be employed in an unsupervised parameter-efficient finetuning of the VL model on a set of unlabeled images. Extensive empirical evaluations show the benefits of our approach w.r.t. UPL and CLIP-PR.

**Prompt Tuning.** Prompt tuning belongs to a wider family of pararamter-efficient finetuning methods that can also be applied to VL models [33, 34]. Originating in NLP [35, 36], visual prompt tuning was recently shown to be effective also for vision backbones [11, 33, 37]. CoOp [33] learns prompt vectors by minimizing the prediction error using the cross-entropy loss. ProDA [37] learns diverse prompts from data to cope with variance of the visual representations. UPL [34] proposes an unsupervised prompt learning framework. TPT [38] proposes a test-time prompt tuning framework. CLIP-Adapter [39] and Tip-Adapter [40] employ an alternative parameter efficient finetuning strategy using additional adapter modules. Recently, CoCoOp [41] proposed a Meta-Net for generating image-adaptive prompts for improved generalization to distribution shifts. In this work, we employ parameter-efficient Visual Prompt Tuning (VPT) [11] as the means for adapting the visual encoder of the VL model, but in contrast with other works we do not use any supervised data. Instead, we use a classifier, trained in the text domain using texts automatically generated by an LLM from the set of target classes, to generate pseudo-labels for an unlabeled image collection as a form of self-supervision. Furthermore, we propose to combine VPT with tuning the scale and shift parameters of normalization layers, previously proposed for handling domain shifts [42–44], but to the best of our knowledge never before used to tune VL models.

**Pseudo-labeling.** Pseudo labeling, also known as self-training, is commonly used as a semi-supervised learning technique. Popularized in the seminal works of [45, 46], pseudo-labeling was extended to utilize consistency regularization [47] and augmentation [48]. The pseudo-labeling pipeline of our method is inspired by FixMatch [10] that combined consistency regularization with confidence-based filtering, surpassing SOTA semi-supervised techniques at the time. It was later extended with a non-parametric classifier in PAWS [49]. These techniques are proposed for semi-supervised learning and require some amount of labeled instances. In contrast, we propose a label-free method for improving VL models performance on a set of target classes. To achieve this, our method finetunes VL models in a parameter-efficient manner by generating pseudo labels through a text-only classifier trained on a corpus of text data generated by prompting language models.

## 3   LaFTer

CLIP [1] consists of a vision encoder and a text encoder, which project images and texts to a common embedding space. It has been trained on a very large volume (400M) of image-text pairs to align the embedding of each image to the embedding of its corresponding text, at the same time pushing it away from the embeddings of unrelated texts corresponding to other images. In [1], it was demonstrated that the text and vision encoders enable effective zero-shot image classification. Given the set of class

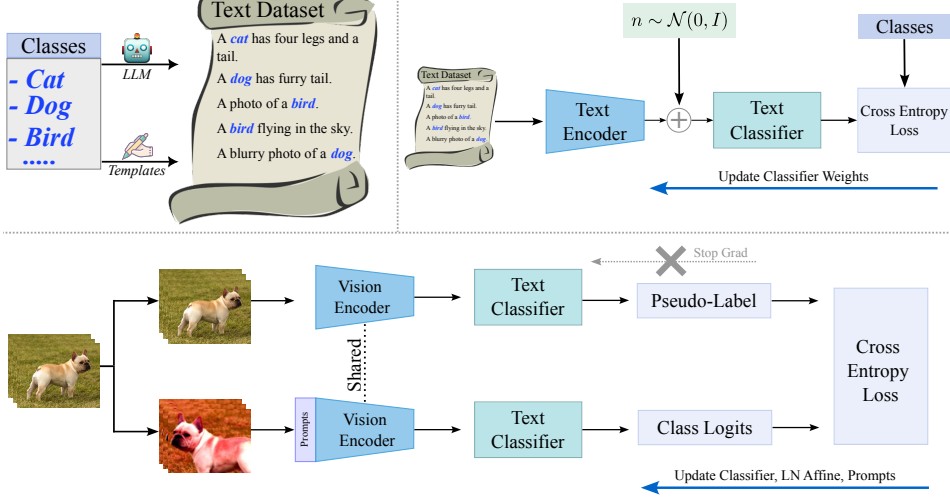

Figure 2: Overview of our LaFTer. (top) Given a set of class labels, we generate a data set of short texts by prompting a Large Language Model (LLM) multiple times with each class name. We compute embeddings of these texts using CLIP text encoder. This lets us train a neural network, the *Text Classifier*, to infer the class used to prompt the LLM from the embedding of the text it generated. Even though the *Text Classifier* has been trained exclusively on text, it performs well in classifying image embeddings generated by CLIP vision encoder. (bottom) We further take advantage of the *Text Classifier* by leveraging it in a pseudo-labeling setup to finetune the VL model.

names $C$, the text encoder $u$ is used to generate the embedding $u_c$ of a prompt for each class $c \in C$, typically of the form '*A photo of ...*', completed with the class name. Each test image $x$ is encoded by the vision encoder $v$ and classified using its cosine similarity $\cos$ to the text embedding of each class. The likelihood of a predicted class $\hat{c}$ is computed with a softmax,

$$l_{\hat{c}}(x) = \frac{e^{\cos(u_{\hat{c}}, v(x))/\tau}}{\sum_{c \in C} e^{\cos(u_c, v(x))/\tau}}, \tag{1}$$

where $\tau$ denotes the temperature constant.

The zero-shot classifier (1) does not require any training data, but is typically outperformed by networks trained on data from the target domain. Existing approaches to reducing this performance gap [30, 31] rely on fine-tuning the zero-shot classifier in a few-shot regime. While effectively enhancing accuracy, they incur additional costs of curating and annotating the training data. In contrast, we propose a two-step approach for fine-tuning the network without requiring any annotations. In the next section 3.1, we describe a technique to train a visual classifier on purely textual data, conveniently generated by Large Language Models. In Section. 3.2, we propose an unsupervised training setup that lets us benefit from unlabelled data from the target domain, when such data is available. Combining these techniques significantly reduces the performance gap to the supervised classifier and is even competitive with few-shot approaches despite not using any supervision. These results are provided in Section 4.2.

## 3.1 Learning an Image Classifier using Text

Aligning the text and image embedding spaces is the key idea underlying Vision Language models. It gives CLIP and similar methods, their main advantages: the capacity to be trained on an extremely large volume of data available on the Internet and the resulting effectiveness as zero-shot classifiers. Recently, Zhang et al. [50] explore yet another advantage of the alignment between text and image embedding spaces - it allows diagnosing and rectifying vision models spawn from the VL model vision encoder by using the other modality. In particular, it hints that it might be possible to finetune a zero-shot image classifier on textual data.

While acquiring and annotating images requires manual labor, a training set of annotated texts can be constructed automatically, using a Large Language Model, like the GPT-3 [9]. Generating text using an LLM constitutes a convenient alternative to text mining, as LLMs represent extremely

large text corpora on which they were trained. Moreover, prompting LLMs is much more efficient than searching a large body of text for the words of interest. To construct our training set, we take inspiration from [51] and for each class $c$, we prompted GPT-3 with queries of the following kind: '*Describe what a [c] looks like.*' We repeated the prompting for each class $c$, in the dataset and complemented the resulting set of synthetic texts with text generated procedurally, using the same hand-crafted templates as for constructing prompts for the zero-shot classifier, for example, '*A photo of a [c].*'. We defer the full list of prompts (to LLMs) and templates to the supplementary material.

The capacity to finetune the classifier in a supervised text classification setting lifts the architectural constraints imposed by the zero-shot setup. More precisely, the class likelihoods no longer need to be produced according to (1), but can instead be computed by a trainable classifier $f$, of arbitrary architecture. We train $f$ with the Smoothed Cross Entropy loss $\mathcal{L}_{\text{SCE}}$ [52]. To regularize training, we add Gaussian Noise $n \sim \mathcal{N}(0, I)$ to the $l_2$-normalized feature vector from the clip text encoder. Formally, given a text training set $T$ consisted of pairs of text fragments $t$ and class labels $c$, the training objective is:

$$\min_\theta \sum_{\substack{(t,c) \in T \\ n \sim \mathcal{N}(0,I)}} \mathcal{L}_{\text{SCE}}\big(f_\theta\big(\frac{u(t)}{\|u(t)\|} + n\big), c\big), \tag{2}$$

where $\theta$ is the parameter of the classifier. Training of the text-only classifier is very efficient. For example, 3000 epochs of training the classifier on the data set of 130000 text sentences, representing the 1000 classes of the ImageNet [53] dataset is completed in $\sim 120$ seconds on an NVIDIA 3090 graphics card.

## 3.2 Unsupervised Finetuning on Target Domain Images

Text-only training does not require any image data. However, in many applications, unlabeled images from the target domain are readily available, or can be acquired at a low cost. Given a set of unlabeled images we propose to take advantage of the text-only pre-trained classifier and use it in a pseudo-labeling pipeline on top of the vision encoder as demonstrated at the bottom part of Figure 2.

Inspired by Fixmatch [10], for each training image $x$, we generate two views: the weakly-augmented view $\alpha_w(x)$ and the strongly-augmented view $\alpha_h(x)$, where $\alpha$ denotes a stochastic augmentation function. Contrary to Fixmatch, in our unsupervised finetuning we set $\alpha_w$ as an identity transformation. For $\alpha_h$ we use the augmentations proposed in [54]. The weakly-augmented view serves to generate a pseudo-label. To that end, it is passed through the vision encoder $v$ and the text classifier $f$, yielding a vector of class probabilities $p$. Class probabilities give rise to pseudo labels, that we denote by $\hat{c}(x)$. Formally,

$$\hat{c}(x) = \arg\max_{c \in C} p_c, \qquad \text{where} \qquad p = f\Big(v\big(\alpha_w(x)\big)\Big). \tag{3}$$

$\hat{c}(x)$ is not differentiable, and we do not backpropagate the error signal through it during training. The pseudo labels $p_{\hat{c}}$, are used as ground truth for training the network on the heavily-augmented views. Since the vision encoders in the two branches share the weights, the pseudo-labels are generated in an *online* manner and are constantly refined as adaptation is progressing. This is in contrast to UPL [12] and CLIP-PR [32] which rely on *offline* pseudo-labeling by only generating them once from frozen CLIP encoders. We freeze the weights of the text-only pre-trained classifier in the pseudo-labeling branch since it leads to more stable adaptation.

When processing the heavily augmented views $\alpha_h(x)$, we augment the vision encoder by visual prompt tuning [11]. That is, we append randomly initialized, learnable parameters (a matrix of size $N_P \times d_v$ where $N_P$ is the number of prompts and $d_v$ is the channel dimension of the vision encoder) to the input of the vision transformer after the initial embedding layer. This helps the network account for the heavy augmentation of the input and, as we show in Section 4.3, boosts the performance of the finetuned classifier. We denote the vision encoder with prompting by $v^p$. The prediction for the heavily-augmented image is obtained as $f(v^p(\alpha_h(x)))$.

The network is trained with the smoothed cross entropy loss $\mathcal{L}_{\text{SCE}}$ [52]. We denote the set of unlabelled target domain images by $D$, and formalize the training objective as

$$\min_{\theta, \eta} \sum_{x \in D} \mathcal{L}_{\text{SCE}}\Big(f_\theta\Big(v_\eta^p\big(\alpha_h(x)\big)\Big), \hat{c}(x)\Big), \tag{4}$$

| | ImageNet | CIFAR-10 | CIFAR-100 | EuroSat | DTD | CALTECH-101 |
|---|---|---|---|---|---|---|
| CLIP | 61.9 | 88.8 | 64.2 | 45.1 | 42.9 | 90.5 |
| CLIP-PR | 60.4 | 89.3 | 63.2 | 44.2 | 40.1 | 84.8 |
| UPL | 61.2 | 89.2 | 65.8 | 62.2 | **48.0** | 90.6 |
| LaFTer | **64.2** | **95.8** | **74.6** | **73.9** | 46.1 | **93.3** |
| | UCF-101 | Flowers-102 | SUN-397 | ImageNet-A | ImageNet-S | ImageNet-R |
| CLIP | 61.0 | 66.6 | 60.8 | 29.6 | 40.6 | 65.8 |
| CLIP-PR | 57.9 | 57.7 | 54.7 | 11.6 | 38.6 | 54.1 |
| UPL | 63.9 | **71.5** | **66.0** | 26.9 | 42.4 | 65.6 |
| LaFTer | **68.2** | 71.0 | 64.5 | **31.5** | **42.7** | **72.6** |

Table 1: Top-1 Classification Accuracy (%) while using the CLIP pre-trained ViT-B/32 backbone for 12 image classification benchmarks. LaFTer represents results obtained by first pre-training the visual classifier on text-only data and then performing unsupervised finetuning on the unlabeled image data. Highest accuracy is shown in bold, while second best is underlined.

where $\theta$ and $\eta$ denote the finetuned parameters of the classifier and of the vision encoder, respectively. The parameters $\eta$ are the visual prompts and the scale and shift (affine) parameters of the normalization layers of the vision encoder. The selection of $\eta$ is motivated by keeping the adaptation *parameter-efficient*. For LaFTer, the number of trainable parameters are less than $0.4\%$ of the entire model parameters, making adaptation extremely lightweight.

## 4 Experimental Evaluation

Here we first provide a description of the datasets and baselines we use in our evaluations, then explain our implementation details and later discuss our experimental results in detail.

### 4.1 Evaluation Setting

**Datasets:** We extensively evaluate our approach on 12 different datasets belonging to widely different domains. More specifically, we use four datasets containing common natural categories: ImageNet [53], CIFAR-10/100 [55] and Caltech-101 [56]. EuroSat [57] contains satellite images of 10 different locations. UCF-101 [58] is an action recognition dataset. SUN-397 [59] contains images from 397 naturally occuring scenes. Flowers-102 [60] is a fine-grained classification dataset for classifying different categories of flowers commonly occuring in the United Kingdom. Whereas, ImageNet-A (Adversarial) [61], ImageNet-S (Sketch) [62] and ImageNet-R (Rendition) [63] are different versions of the original ImageNet validation set. In our setting, we divide the ImageNet-A, ImageNet-S and ImageNet-R in to 80% train and 20% test set. For all other datasets we use the splits provided by [30].

**Baselines:** We compare LaFTer with baselines which are label-free (not requiring any additional labeled images):

- **CLIP** [1] denotes zero-shot classification scores by computing the cosine similarity between embeddings from frozen CLIP encoders.
- **UPL** [12] adds learnable text prompts to the CLIP text encoder and finetunes them in an unsupervised manner by employing confidence-sampled offline pseudo-labeling.
- **CLIP-PR** [32] optimizes an adapter on top of the CLIP vision encoder by using label distribution priors from the training set of the downstream datasets and generating offline pseudo-labels.

For completeness, apart from these 3 baselines, we also provide a comparison with few-shot fine-tuning method CoOp [30], which learns *soft* text prompts using $k$ labeled images per class ($k$-shot).

**Implementation Details:** For all our experiments, unless otherwise stated, we use a ViT/B-32 CLIP pre-trained model from OpenAI [1]. The Text-Only classifier (Section 3.1) is implemented as a

| | ImageNet | CIFAR-10 | CIFAR-100 | EuroSat | DTD | CALTECH-101 |
|---|---|---|---|---|---|---|
| LaFTer (no-shot) | 64.2 | 95.8 | 74.6 | 73.9 | 46.1 | 93.3 |
| CoOp (1-shot) | 60.6 | 83.0 | 55.6 | 58.4 | 40.1 | 91.7 |
| CoOp (5-shot) | 61.3 | 86.6 | 63.2 | 71.8 | 41.1 | 93.2 |
| CoOp (10-shot) | 62.3 | 88.5 | 66.6 | 81.6 | 65.8 | 94.6 |
| PEFT (1-shot) | 50.7 | 62.7 | 50.2 | 37.5 | 42.6 | 90.6 |
| PEFT (5-shot) | 59.3 | 80.0 | 67.3 | 55.3 | 59.9 | 94.5 |
| PEFT (10-shot) | 62.8 | 87.9 | 74.1 | 67.9 | 67.3 | 96.1 |
| | UCF-101 | Flowers-102 | SUN-397 | ImageNet-A | ImageNet-S | ImageNet-R |
| LaFTer (no-shot) | 68.2 | 71.0 | 64.5 | 31.5 | 42.7 | 72.6 |
| CoOp (1-shot) | 63.8 | 71.2 | 64.1 | 24.5 | 39.9 | 60.0 |
| CoOp (5-shot) | 74.3 | 85.8 | 67.3 | 30.0 | 46.5 | 61.6 |
| CoOp (10-shot) | 77.2 | 92.1 | 69.0 | 35.0 | 49.1 | 63.6 |
| PEFT (1-shot) | 60.5 | 66.9 | 58.3 | 20.9 | 38.5 | 57.2 |
| PEFT (5-shot) | 72.6 | 91.1 | 68.7 | 33.3 | 55.3 | 66.4 |
| PEFT (10-shot) | 79.8 | 95.2 | 72.3 | 40.2 | 61.1 | 71.0 |

Table 2: Top-1 Accuracy (%) for our LaFTer (no-shot) compared to few-shot methods. We compare to CoOp [30] in 1-, 5- and 10-shot supervised finetuning regimes. Parameter Efficient Finetuning (*PEFT*) represents tuning the same parameters as in LaFTer (prompts, classifier, affine) but in a few-shot manner. For each dataset/compared method, blue highlights the highest number of shots outperformed by *no-shot* LaFTer. Notably, LaFTer improves over 10-shot and all compared methods in 4 datasets, including ImageNet, where 10-shot = 10K labeled samples.

single linear layer, with the output units equal to the number of classes in the dataset. For training this classifier, we load the complete text dataset as a single batch and optimize the network using AdamW as optimizer, with a learning rate of $0.001$. For unsupervised fine-tuning using visual data (Section 3.2), we again use the AdamW optimizer with a learning rate of $0.0001$, batch size of $50$ and optimize the learnable parameters for a total of $50$ epochs. Thanks to our Text-Only classifier pre-training, through empirical evaluations we find that $10$ epochs are sufficient for fine-tuning on the large-scale ImageNet dataset. To produce an augmented view of the image, we employ the augmentations used in SimSiam [54]: Gaussian blur, random resized crop, random horizontal flip, color jitter and random gray scaling. For generating class descriptions we use different LLM's, e.g., GPT-3 [9] and Alpaca [64]. In total we generate $50$ descriptions for each category in the dataset. We ablate the LLM choice in section 4.3. To construct the text dataset we combine the class descriptions from LLM's and dataset-specific prompt templates provided by [1].

## 4.2 Results

We test our LaFTer extensively on $12$ image classification datasets. These results are provided in Table 1. Our LaFTer consistently improves the zero-shot CLIP model on all the $12$ datasets. On some datasets, for example EuroSat, our LaFTer shows an absolute improvement of over 28% on the zero-shot CLIP classification results. Even on the large scale ImageNet dataset we show a considerable improvement of $2.3\%$ over the zero-shot CLIP classification results.

We also see that we outperform CLIP-PR on all datasets. Since CLIP-PR relies on offline pseudo-labels, which are generated only once for the entire dataset and only proposes to finetune an adapter on top of frozen CLIP visual encoder. We conjecture that their approach might be less expressive. Moreover, it requires label distribution prior from the dataset on which it is being finetuned on. Our LaFTer is free from these requirements and proposes a more general solution for unsupervised finetuning on downstream datasets.

We also compare to the other unsupervised adaptation baseline UPL, which relies on finetuning learnable text prompts attached to the CLIP text encoder through knowledge distillation. In Table 1, we see that our method out performs UPL on most of the datasets ($9$ out of $12$). On some datasets such as EuroSat, it shows a huge gain of $11.7$ percentage-points over UPL. On datasets such as Describable Textures Dataset (DTD), Flowers-102 and SUN-397 our method is marginally behind

| | IN | CIFAR-10 | CIFAR-100 | IN-A | IN-S | IN-R | Mean |
|---|---|---|---|---|---|---|---|
| Class Name | 58.4 | 87.1 | 59.0 | 30.7 | 37.7 | 65.5 | 56.4 |
| Simple-Template | 61.1 | 88.1 | 63.1 | 30.4 | 40.3 | 65.9 | 58.1 |
| Dataset-Templates | 60.1 | 89.3 | 64.7 | 31.1 | 41.2 | 67.5 | 59.0 |
| Alpaca Descriptions | 54.8 | 88.4 | 59.4 | 25.7 | 36.3 | 58.7 | 53.9 |
| GPT Descriptions | 60.5 | 87.8 | 63.0 | 30.2 | 39.6 | 63.6 | 57.4 |
| GPT + Templates | 61.9 | 89.3 | 65.4 | 31.5 | 40.9 | 67.8 | 59.5 |

Table 3: Top-1 Classification Accuracy (%) for a ViT-B/32 model while ablating different text generation strategies for training the text-only classifier in the first stage of LaFTer . To obtain these results, we evaluate the test set of the respective datasets by using the text-only pre-trained classifier on top of the frozen vision encoder from CLIP. IN = ImageNet.

| | Clip | w/o Aug | w/o Prompts | w/o Affine | w/o Cls | w/o Stop Grad | w/o Text Pre-train | LaFTer |
|---|---|---|---|---|---|---|---|---|
| CIFAR-10 | 88.8 | 93.5 | 92.5 | 94.6 | 94.1 | 94.7 | 95.1 | 95.8 |
| CIFAR-100 | 64.2 | 72.6 | 71.7 | 72.9 | 67.4 | 73.2 | 73.1 | 74.6 |
| UCF-101 | 61.0 | 65.3 | 66.1 | 63.6 | 63.4 | 67.5 | 67.7 | 68.2 |
| EuroSat | 45.1 | 63.2 | 61.2 | 64.2 | 60.9 | 69.2 | 69.7 | 73.9 |
| ImageNet-A | 29.6 | 29.9 | 29.8 | 30.8 | 30.1 | 31.1 | 28.9 | 31.5 |
| ImageNet-S | 40.6 | 40.3 | 40.7 | 41.1 | 40.9 | 42.1 | 41.3 | 42.7 |
| ImageNet-R | 65.8 | 68.0 | 67.7 | 66.8 | 66.1 | 71.8 | 70.2 | 72.6 |
| **Average** | 56.4 | 61.8 | 61.4 | 62.0 | 60.4 | 64.2 | 63.7 | 65.6 |

Table 4: Top-1 Accuracy (%) for our LaFTer while ablating the various critical design choices in our methodology. For each of these experiments, we disable one component from our framework and test the resulting method. Aug: Augmentations, Cls: Classifier.

UPL. We conjecture that since we use augmentations in one of our streams during our adaptation phase, it might result in noisy gradients during the learning process. Specially since datasets like DTD and Flowers-102 can depend on color cues for distinguishing the classes. For the large scale ImageNet dataset, we see that the performance of UPL is below the CLIP zero-shot performance. This can be because UPL generates offline pseudo-labels and in ImageNet, due to fine-grained classes the pseudo-labels might not be very confident from the zero-shot CLIP classifier. On the other hand, LaFTer benefits first from a classifier which has learned discriminative visual features through text-only training and later makes use of parameter efficient finetuning (PEFT). Furthermore, since our pseudo labels are generated in an online manner for each iteration during the optimization, they are also constantly refined as the training is progressing.

In Table 2 we provide a comparison of our LaFTer with the few-shot learning method CoOp [30] and also test Parameter Efficient Fine-Tuning (PEFT), which tunes the same learnable parameters as LaFTer (prompts, classifier, and affine parameters of the normalization layers), but in a few-shot manner. Interestingly, we see that our unsupervised representation learning method conveniently outperforms CoOp for 1- and 5-shots. For example, LaFTer (*no-shots)* is on average 7.1% better than CoOp (1-shot) and even 1.3% better in the 5-shot learning regime, while remaining competitive for 10-shots. It is also worth noting that for the large-scale ImageNet our LaFTer (requiring no labels) performs better than CoOp (10-shots), requiring 10000 labeled instances from the dataset. Results also follow a similar trend when compared with PEFT.

## 4.3 Ablation Studies

To understand the significance of all the components in our LaFTer we minutely study each design aspect. First we discuss the performance of our Text-Only pre-trained classifier, then we ablate each component in our unsupervised adaptation phase using unlabeled images and finally provide results by adapting other pre-trained backbones from CLIP [1]. Due to limited evaluation resources and space constraints, we perform these ablations on a subset of datasets with different complexity.

**Text-Only Pre-trained Classifier.** A main component of our LaFTer is the text-only pre-trained visual classifier, later used in our pseudo-labeling pipeline. The motivation behind it being, that since

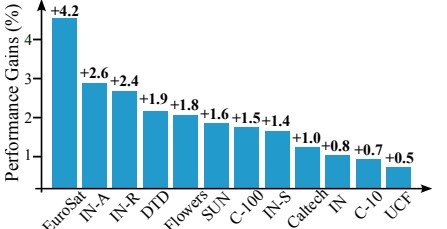

| | C-10 | C-100 | UCF | EuroSat | IN-R |
|---|---|---|---|---|---|
| CLIP (B/16) | 89.2 | 68.1 | 64.7 | 48.4 | 73.8 |
| LaFTer | 96.5 | 76.3 | 67.2 | 72.1 | 81.4 |
| CLIP (L/14) | 95.3 | 75.8 | 72.0 | 60.3 | 84.9 |
| LaFTer | 99.0 | 87.2 | 77.2 | 77.2 | 91.5 |

Figure 3: Performance gains of using the text-only pre-trained visual classifier vs not using it in LaFTer pseudo-labeling pipeline scheme.

Table 5: Top-1 Accuracy (%) for CLIP pre-trained ViT-B/16 and ViT-L/14 backbones. Results are provided for Base VL model (CLIP) and our LaFTer.

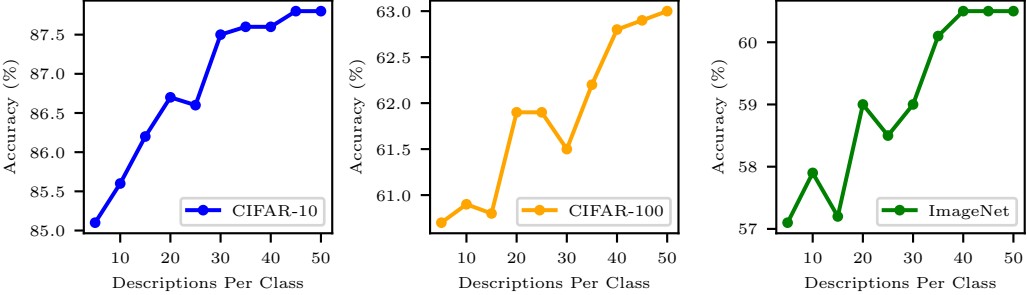

Figure 4: Effect of diversity of descriptions on the Top-1 Accuracy (%). We train the text classifier by randomly choosing (with an increment of 5) a certain number of descriptions per class for each evaluation step. For all the main evaluations, we use a maximum of 50 descriptions per class.

CLIP is trained in order to have a shared text and vision embedding space so a classifier trained to classify the embeddings from any one of the modalities should also be able to classify embeddings from the other modality. We show that it is possible to train a classifier to classify images by only training it to classify natural language. To design this self-supervised objective of classifying text (described in detail in 3.1) we test different ways of generating the text dataset. For example, simple-template such as *A photo of a ...*, dataset-specific templates from CLIP [1] and descriptions from LLMs. These results are presented in Table 3. Note that for these results, we first train the classifier on the generated text dataset and then evaluate on the (images) test set of the respective dataset by using the trained classifier to classify the visual embeddings from the CLIP visual encoder. The simplest text dataset generation strategy: classifying the *classname* is also able to show reasonably strong visual classification performance. Furthermore, different strategies have slight difference in performance. We also find that descriptions from GPT-3 work better than descriptions from Alpaca. Our choice of generating class descriptions by prompting GPT-3 and complementing them by adding handcrafted templates from [1] works best. While averaging over 6 datasets, we gain a performance improvement of 3.1% in comparison to the simplest text dataset generation strategy of classifying the classnames themselves.

**Text Classifier Contribution to LaFTer.** Pre-training a visual classifier on text-only data and then employing it in our pseudo-labeling pipeline helps LaFTer gain improvements on all the 12 datasets which we evaluate in our paper. In Figure 3 we analyze the performance gains for our LaFTer when using the text-only pre-training in conjunction with our pseudo-labeling pipeline. On some datasets, for example EuroSat [57], our text-only pre-training helps our LaFTer to gain an absolute performance improvement of 4.2%. On other datasets, the performance gains are also consistent.

**Design Choices for Unsupervised Finetuning.** In order to study the effect on performance of each individual component we ablate our LaFTer in Table 4. We see that removing the classifier and instead using the CLIP Cosine similarity scores in both branches results in the highest degradation of results. Similarly, all other design choices also have a certain effect. For example, removing the learnable visual prompts results in a decrease of 4.2 percentage-points as compared to our LaFTer.

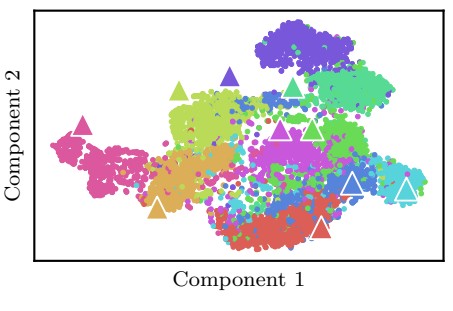
(a) Base CLIP (ViT-B/32).

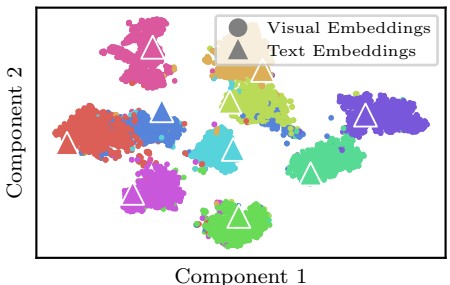
(b) LaFTer Adapted CLIP (ViT-B/32).

Figure 5: TSNE projections for visual *(circles)* and text *(triangles)* embeddings for 10 classes of the EuroSAT dataset from (a) Base *(un-adapted)* CLIP ViT-B/32 model and (b) after adaptation with LaFTer. For the Base CLIP model, the text embeddings are the output feature vector from the CLIP text encoder and for LaFTer we use the weights of the adapted text-classifier. Visual embeddings are always the output features from the vision encoder.

**Different CLIP Backbones.** For our main experimental results described in Tables 1 and 2 and ablations in Tables 3 and 4, we use the ViT-B/32 CLIP backbone. In Table 5 we also provide results with different backbones for our LaFTer . We observe consistent improvements by LaFTer over CLIP also for larger backbones. For example, for ViT-B/16 our method shows 23.7% absolute improvement for EuroSat, while for ViT-L/14, our method improves CLIP zero-shot by 16.9% on the same dataset.

**Diversity of Descriptions.** To test the effect of diversity of generated descriptions on the visual classification results for the text-only pre-training of the text-classifier, we ablate the number of descriptions chosen for each class (for ImageNet and CIFAR-10/100) and provide the results in Figure 4. We find that increasing the diversity of descriptions to generate the text dataset has a positive effect for all the 3 datasets ablated.

**Visualization of Embeddings.** To gain further insights into the LaFTer adaptation we visualize the latent embeddings from the CLIP encoders in Figure 5, before and after adaptation for all 10 classes in the EuroSAT dataset. Analyzing the TSNE projections, we observe that the adaptation with our LaFTer results in more pronounced category clusters with larger inter-class distances, and also that after applying LaFTer the class-label text embeddings are better aligned with these category clusters.

## 5   Conclusion & Limitations

We propose a completely label-free finetuning method for Vision-Language models by first showing cross-modality transfer and learning a classifier on natural language inputs which can successfully classify visual data. Later, we leverage this text-only pre-trained classifier in our pseudo-labeling pipeline to further finetune the VL models in a parameter efficient manner. We extensively evaluate our LaFTer and achieve state-of-the-art results when comparing to other methods in an unsupervised finetuning paradigm, while also performing favorably in comparison to methods relying on few-shot supervised learning routines.

**Limitations.** One contribution of LaFTer is to show cross-modality transfer by training a classifier to classify natural language sentences and showing that it can be readily applied to classify visual data as well. In our work, we experimented with a single linear layer as a classifier. Due to the risk of over-fitting because of the sparse nature of natural language, we did not experiment with more complex structures. Experimentation with expanding the text dataset coupled with designing a more expressive classifier head is left as an exciting future direction.

## Acknowledgements

We gratefully acknowledge the financial support by the Austrian Federal Ministry for Digital and Economic Affairs, the National Foundation for Research, Technology and Development and Christian Doppler Research Association. This work was also partially funded by the FWF Austrian Science

Fund Lise Meitner grant (M3374) and the Austrian Research Promotion Agency (FFG) under the SAFER (894164) project.

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
