# LaFTer: Label-Free Tuning of Zero-shot Classifier using Language and Unlabeled Image Collections Supplementary Material

**M. Jehanzeb Mirza**[1,2]     **Leonid Karlinsky**[3]     **Wei Lin**[1]     **Mateusz Kozinski**[1]
**Horst Possegger**[1]     **Rogerio Feris**[3]     **Horst Bischof**[1,2]

[1]Institute of Computer Graphics and Vision, TU Graz, Austria.
[2]Christian Doppler Laboratory for Embedded Machine Learning.
[3]MIT-IBM Watson AI Lab, USA.

Project Page: https://jmiemirza.github.io/LaFTer/

In the following, we first list the implementation and computation details. Then, provide experiments performed in a more challenging scenario where the unlabeled training data can have unrelated (noisy) samples from unrelated classes. Finally, we provide details on the construction of the text dataset for training a visual classifier on text.

## 1  Implementation and Computation Details

We implement our LaFTer in the PyTorch framework and in order to promote reproducibility provide the code as supplementary material. The codebase contains a detailed `Readme` file, which explains how to set up the datasets and run experiments for LaFTer. For running experiments for our LaFTer and all the baselines we used a single GPU cluster consisting of $4$ NVIDIA Quadro Graphic Cards. To run all experiments for CLIPPR [1] and UPL [2] we use the official codebase released by the respective authors[1,2]. Please note, CLIPPR used the same CLIP ViT-B/32 backbone, while UPL used weaker backbones, so we evaluated their approach for the CLIP pre-trained ViT-B/32 backbone for a more fair comparison.

## 2  Unrelated Samples during Adaptation

In real-world applications, the unlabeled image collection, such as the one we use in LaFTer (in conjunction with the text-only training) can also contain unrelated images, e.g., images of other classes, not belonging to the target classes set. An unsupervised adaptation method should ideally be robust against such outliers in the adaptation phase. We test our LaFTer and other baselines in 2 such scenarios, described as follows:

**Unrelated Samples from Other Datasets:**   To evaluate this scenario, we add unrelated class samples to the unlabeled CIFAR-10 training experiment from the main paper. Specifically, we add to the unlabeled set of all CIFAR-10 images additional (unlabeled) 'noise' images from $N$ classes ($N =$ 10, 20, ..., 90) of CIFAR-100 that do not overlap CIFAR-10 classes. We tune LaFTer and baselines on this noisy unlabeled set and evaluate the resulting models on the same CIFAR-10 test set (keeping the target classes to be only the CIFAR-10 classes). We plot the results obtained in this scenario for our LaFTer and other baselines in Figure 1. We see that our LaFTer is robust to adding unrelated classes during the adaptation phase. As we can see, there is less than $2\%$ of a performance drop when adding all the 90 classes (as unrelated samples) from CIFAR-100 during adaptation on CIFAR-10 as compared to adding no noise classes from CIFAR-100. We also observe that the baselines mostly

---

[1]CLIPPR: https://github.com/jonkahana/CLIPPR, Commit: 96c1f23
[2]UPL: https://github.com/tonyhuang2022/UPL, Commit: 97f671f

37th Conference on Neural Information Processing Systems (NeurIPS 2023).

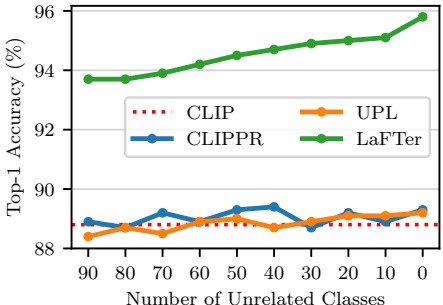

Figure 1: Top-1 Accuracy (%) for CIFAR-10 dataset (with ViT-B/32 backbone) while having **unlabeled** samples of *unrelated* classes from CIFAR-100 dataset added to the **unlabeled** CIFAR-10 set while keeping the target classes set to be CIFAR-10 classes only.

|  | EuroSat | CIFAR-100 |
|---|---|---|
| CLIP | 55.0 | 73.1 |
| CLIPPR | 56.1 | 73.2 |
| UPL | 71.3 | 75.0 |
| LaFTer | **77.6** | **80.6** |

Table 1: Top-1 Accuracy (%) for ViT-B/32 backbone when using only 50% of the classes as target classes for evaluation and having unlabeled samples from all other classes as unrelated (noise) data in the unlabeled image collection used for the adaptation.

under-perform their source CLIP model (that has 88.8% zero-shot accuracy on CIFAR-10 without tuning) hence neither improving nor deteriorating the performance on the noisy unlabeled set.

**Unrelated Samples from Same Dataset:** In order to test our adaptation method in the presence of more fine-grained unlabeled noise samples in the unlabeled image collection used for the adaptation, we simulate a scenario where we treat samples from 50% of the classes from the same dataset as unrelated samples while using the remaining 50% of the classes as target classes for the adaptation and evaluation. These results are provided in Table 1. For EuroSat, in this scenario, our text-only classifier is only trained to classify 5 classes from the dataset for a closed-set classification scenario (while the other 5 classes are not revealed). However, during the (second) unsupervised adaptation phase, we also add unlabeled samples from the remaining 5 classes in the unlabeled image collection to serve as unrelated (out-of-distribution noise samples). We see that our LaFTer shows strong performance gains also in such a challenging more fine-grained noise scenario and also performs better than other baselines. Results for CIFAR-100 in this scenario, also follow a similar trend.

## 3 Text Dataset

In the first step of our LaFTer we propose to train a visual classifier on a dataset consisting of natural language, showing successful cross-modal transfer. To build this dataset we try different methods, which are ablated in the main manuscript (Table 3). Due to the better performance obtained by mixing the descriptions from GPT-3 [3] and dataset-specific templates, we choose this method of designing the text dataset. In the following, we first provide the list of queries (prompts) we use to obtain descriptions from GPT-3, then provide some qualitative examples of descriptions and finally provide some examples of dataset-specific templates adopted from [4].

### 3.1 List of Prompts

We follow [5] and query GPT-3 with different prompts in order to obtain descriptions for each class. In total, we use 5 prompts and require the LLM to generate 10 responses for each prompt. The list of these prompts is as follows:

- Describe what a category looks like.

- How can you identify a category?

- What does a category look like?

- Describe an image from the internet of a category.

- A caption of an image of a category?

Here, category is replaced by the actual classname from the dataset and the response from the LLM is automatically matched with the true classname.

## 3.2 Qualitative Examples

By querying the LLM for descriptions, we can potentially generate a huge corpus of text samples representing each class. Some examples of the responses from the LLM, when we query it with the prompts mentioned above for the class quail from the ImageNet [6] dataset, include:

- A quail is a small game bird with a rounded body and a small head.
- A quail is a small, plump bird with a round body and a short tail.
- A quail can be identified by its plump body, short legs, and small head with a pointed beak.
- A quail can be identified by its small, rounded body and short tail.
- A quail looks like a small chicken.
- A quail is a small, crested game bird.
- This image is of a quail in a natural setting.
- In the image, there is a brown and white quail perched on a branch.
- A quail hiding in some foliage.
- A young quail pecks at the ground in search of food.

## 3.3 Dataset Specific Templates

We complement the descriptions from the LLM with the dataset specific templates provided by [4], to obtain our text dataset for training the visual classifier. For example, for the ImageNet dataset, we find that the following 7 templates work best for training the classifier (results obtained by using different types of templates and data generation strategies are listed in Table 3, main manuscript):

- a bad photo of the category.
- a category in a video game.
- a origami category.
- a photo of the small category.
- art of the category.
- a photo of the large category.
- itap of a category.