# OpenReview forum: "LaFTer: Label-Free Tuning of Zero-shot Classifier using Language and Unlabeled Image Collections"
_NeurIPS.cc/2023/Conference — NeurIPS 2023 poster_

### Official Review · Reviewer_o7C2 · 2023-07-04

**Soundness:** 4 excellent
**Presentation:** 4 excellent
**Contribution:** 3 good
**Rating:** 6
**Confidence:** 5

**Summary:**

The paper proposes a novel method for improving zero-shot classification of VL models, in an unsupervised manner. This is done without additional visual labeled examples, yet with additional unlabeled examples. They rely on LLM to generate a dataset of text describing the desired classes. Then, they train a text classifier on top of the VL textual encoder using the generated dataset. As a second stage, they use a set of relevant unsupervised images as well as a set of designed augmentations, to finetune the classifier and visual prompts, adjusting it to the VL models image representations. In an extensive evaluation, the authors show a significant performance enhancement, and ablate their method methodically.

**Strengths:**

-	The paper is clearly written and easy to follow.
-	The performance enhancement shown by LaFTer is very significant and consistent across most test cases.
-	Although inspired by previous works, the idea of tunning the visual encoder of VL models using text supervision is highly interesting.

Overall, I believe the idea of the paper is very interesting, and the results are a significant improvement over CLIP.


**Weaknesses:**

My main concern is the performance of CLIPPR baseline. In most cases, it has been found to be less successful than the original CLIP, which is not intuitive with the method. Could the authors explain this gap please?

One other minor concern is the statement in the abstract about LaFTer being the first to reduce the gap from the supervised baseline. Table 1 shows LaFTer is not the first. Table 2 shows that it does not close the gap for about 50% of the cases, using only few-shot supervision. I believe this is misleading statement.


**Questions:**

-	Could the authors please verify if the experiments denoted by LaFTer* are just the second stage of LaFTer using CLIP as the initial text classifier?
If so, I think these results are more suitable to appear in the ablations section (Table 4).


**Limitations:**

-	The method requires access to a set of unlabeled images from the same distribution.
-	The method is limited by the quality of the visual representation of the VL model.

---

> ### Author Rebuttal · Authors · 2023-08-08
>
> Thank you for the time and effort spent in reviewing our paper. In the following, we provide a response to the questions raised in the review.
>
> **Performance of CLIP-PR baseline.** The CLIP-PR reported image classification results only on CIFAR-10 and ImageNet.
> We used the official codebase provided by the CLIP-PR authors to evaluate their method on all the additional
> datasets reported in our paper. We used the values for all hyperparameters as recommended by the CLIP-PR authors in
> their official release.
>
> We would also like to note that the zero-shot classification accuracy of the original (non-adapted) CLIP baseline
> reported in CLIP-PR is lower than we observed in our experiments and reported in our paper.
> For example, in Table $3$ of the CLIP-PR paper, they report $57.59$% CLIP zero-shot Top-1 Accuracy (%) for
> ImageNet, whereas, we measured CLIP to obtain $61.9$% (Table $1$ of our submitted manuscript) while using the same ViT-B/32 backbone from
> OpenAI as reported to be used by CLIP-PR and observed from their official code.
> Similarly, for CIFAR-10 we obtain zero-shot accuracy of $88.8$% for base CLIP, while CLIP-PR reports $85.17$%.
>
> **Confusion about LaFTer being the first to reduce the gap from the supervised baselines.** Thank you for
> pointing it out, and we agree that the referred statement (in its current form) might lead to confusion.
> However, the purpose was to highlight that LaFTer is the first method to jointly combine two concepts in order to
> effectively finetune Vision-Language models in an unsupervised manner:
>
> - Substituting labeled visual instances of the categories by generating text through Large Language Models (Learning an Image Classifier using Text, Section 3.1).
> - Finetuning the visual encoder in an unsupervised manner by using unlabeled image data (Unsupervised Finetuning on Target Domain Images, Section 3.2).
>
> It is true that CLIP-PR and UPL use unlabeled image data for unsupervised finetuning of Vision-Language models, however,
> LaFTer is the first method to also take advantage of auto-generated text from the Large Language Models which
> substitutes visual instances.
> We will make the referred statement in the abstract concise and more clear in the updated version of the manuscript.
>
> **Move LaFTer$^\*$ results to the ablation section.** Thank you for the suggestion.
> Analyzing the tables again, we agree that it will be more suitable to move LaFTer$^*$ results to the ablations section
> (Table  $4$).
> We will rearrange the results in the updated manuscript.

---

> > ### Comment · Reviewer_o7C2 · 2023-08-17
> >
> > Thank you for your response.
> >
> > CLIPPR Baseline Results:
> > There is a mismatch between the setting of CLIPPR and CLIP. As mentioned in the CLIPPR paper they used just a single prompt, where in CLIP a set of handcrafted prompts was used. I believe the setting of CLIPPR should be adjusted for a fair comparison. This concern remains unaddressed.

---

> > > ### Author Response · Authors · 2023-08-17
> > > **On adding multi-prompt CLIP-PR baseline**
> > >
> > > Thanks for pointing this out! To further address the reviewer's concern with the CLIP-PR single prompt evaluation in Table 1, we have also evaluated CLIP-PR in _multi-prompt_ setting (using the CLIP hand-crafted templates) as suggested by the reviewer. _Multi-prompt_ setting improves CLIP-PR result by 2.9% (averaging over all the evaluated benchmarks presented in Table 1 of our paper). But even in this setting, LaFTer has a 9.0% average advantage over the _multi-prompt_ CLIP-PR. That is, LaFTer average over all the evaluation benchmarks of Table 1 is 9.0% higher than the average of _multi-prompt_ CLIP-PR with positive gains for all the benchmarks (gains ranging between 1% and 22.6%). We provide the detailed results in the table below and will add the _multi-prompt_ CLIP-PR baseline and its results to the revised version of the paper (Table 1), thanks for this suggestion!
> > >
> > > |                  | IN    | C10    | C100   | EuroSat | DTD   | Caltech |
> > > |------------------|-------|--------|--------|---------|-------|---------|
> > > | CLIP-PR (Single) | 60.4   | 89.3      | 63.2   |   44.2   |   40.1 |     84.8   |
> > > | CLIP-PR (Multi)  | 61.1  |   89.5  |    65.3  |    51.3   |   45.1 |     88.7   |
> > > | LaFTer           | 64.2  |   95.8  |    74.6  |    73.9   |   46.1 |     93.3   |
> > > |                  | **UCF**   | **Flower** | **SUN**    | **IN-A**    | **IN-R**  | **IN-S**    |
> > > | CLIP-PR (Single) | 57.9   | 57.7   |    54.7    | 11.6       | 38.6  |   54.1   |
> > > | CLIP-PR (Multi)  | 59.7  |   60.1  |    57.0    | 15.2    |   40.8 |     56.9   |
> > > | LaFTer           | 68.2   | 71      | 64.5   | 31.5    |  42.7 |     72.6   |

---

> > > > ### Comment · Reviewer_o7C2 · 2023-08-18
> > > >
> > > > Thank you for the quick response.
> > > >
> > > > Could the authors please upload a table with detailing the updated CLIPPR results?

---

> > > > > ### Author Response · Authors · 2023-08-18
> > > > > **Detailed multi-prompt CLIP-PR results added**
> > > > >
> > > > > Sure! We edited the response above to include the detailed table in markdown format.
> > > > > Please let us know if you have any further concerns, we would be happy to answer!

---

### Official Review · Reviewer_zXQC · 2023-07-05

**Soundness:** 4 excellent
**Presentation:** 3 good
**Contribution:** 3 good
**Rating:** 6
**Confidence:** 5

**Summary:**

This paper proposes a finetuning approach for Vision-Language models that does not require any labels. It begins by demonstrating the transfer of information between different modalities and training a classifier using natural language inputs. This classifier achieves the successful classification of visual data. Furthermore, it utilizes this pre-trained classifier that relies solely on text inputs in our pseudo-labeling pipeline. This enables it to effectively and efficiently fine-tune the Vision-Language models with fewer parameters.

**Strengths:**

1- The proposal introduces a label-free approach to zero-shot learning, capitalizing on the benefits of pretrained LLM models. Additionally, it enhances the model's performance by finetuning it in an unsupervised manner. This novel approach presents a new direction for zero-shot learning research, eliminating the need for labeled datasets even for seen classes.
2- The proposed method extends its experiments to include a few-shot framework and demonstrates a noteworthy improvement compared to the baseline approaches.
3- The experiments are conducted on twelve (12) diverse datasets, encompassing coarse-grained, fine-grained, and natural scene datasets. A comprehensive ablation study is conducted to demonstrate the effectiveness of the proposed model and the contribution of its individual components.

**Weaknesses:**

1- Upon analyzing Table-1, it is evident that the proposed model exhibits lower performance for the Flower and SUN datasets, both commonly utilized in zero-shot learning and are fine-grained datasets. This observation raises surprise as the proposed model struggles specifically with fine-grained datasets.
2- The visualization of embeddings obtained from both the text encoder and image encoder would provide a clearer understanding of the effectiveness of the pretrained CLIP (LLM) model. Additionally, generating images based on textual descriptions of classes would offer valuable insights into how the proposed model operates within the zero-shot framework.
3- The distinction between the weakly-augmented view and the strongly-augmented view is unclear when it comes to unsupervised fine-tuning. What are the benefits of utilizing the weakly-augmented view for pseudo-labels instead of the strongly-augmented view?

**Questions:**

Please answer all my concerns raised in the weaknesses section.

**Limitations:**

In this paper, the authors acknowledge a limitation of their proposed model. They state that in their current work, they employed a straightforward neural network-based classifier and consider exploring more complex neural networks as part of future work. However, it is worth noting that from my observation of Table-1, the proposed model also faces challenges when applied to fine-grained datasets.

---

> ### Author Rebuttal · Authors · 2023-08-08
>
> Thank you for the time and effort spent in reviewing our paper. In the following, we provide a response to the questions raised in the review.
>
> **Performance on fine-grained datasets.** On datasets like Flower-102 and SUN-397 our method improves the base
> CLIP model by $4.4$% and $3.7$% respectively.
> However, as compared to the other unsupervised finetuning baseline UPL, our LaFTer slightly lags behind
> on Flowers-102 and on SUN-397 (while significantly improving over UPL on average across all tested datasets).
> One potential reason could be the descriptions generated from the LLM for the fine-grained categories of these (fine-grained) datasets.
> To probe into this, we consider the example of Flowers-102 dataset.
> It contains categories such as,`love in the mist, mexican aster, alpine sea holly, ruby-lipped cattleya`,
> which are names of flowers.
> It could be challenging for an LLM to provide descriptions for these categories without having any prior knowledge
> about the _parent category_ of these classes.
> In order to generate more exact descriptions about these finegrained classes, we slightly modify the prompts to the
> LLM (GPT) and add the parent category (flower) in the prompts.
> A couple of examples for the modified prompts are as follows (please refer to the supplementary Section 3.1,
> for the original list of prompts):
>
>
> - Describe what the _flower_ type {category} looks like.
> - How can you identify the _flower_ type {category}?
>
>
> By generating descriptions of the finegrained categories by prompting the LLM with the parent category _flower_ we get
> an improvement of $1.4$% ($71.0$% $\to$ $72.4$%) for our LaFTeR final classification accuracy for this dataset.
> The improved results show a gain of $0.9$% over the UPL baseline (similarly, we gain $0.3$% over UPL for SUN-397), and once again demonstrate the ease with which
> the text-only pre-training proposed in LaFTer can facilitate further customization to downstream tasks and domains
> via small (essentially cost-free) changes in the prompts.
> By adapting the LLM instructions to include the _parent category_, we can easily generate better-targeted
> descriptions for any fine-grained dataset.
> Notice that we also experiment with _targeted prompts_ for the out-of-distribution dataset ImageNet-Rendition
> and show a considerable performance improvement.
> We discussed those results in our response to `Reviewer YPJ1`.
> At this point, we leave the exploration of _targeted prompts_ towards downstream datasets as an interesting and
> seemingly very promising future work direction.
>
> **Visualization of embeddings.** In Figure 2 of the global response, we provide TSNE projections of the
> visual and text embeddings, before and after adaptation for the $10$ classes in the EuroSAT dataset.
> Analyzing the TSNE projections, we observe that the adaptation with our LaFTer
> results in more pronounced category clusters with larger inter-class distances, and also that after applying LaFTer the class-label text
> embeddings are better aligned with these category clusters.
>
>
> **Benefits of weakly-augmented view for pseudo-labels.** In the context of self-supervised learning with pseudo-labels,
> several approaches have introduced the concept of consistency regularization by deriving pseudo-labels from
> weakly-augmented data samples, as demonstrated in works like FixMatch [1].
> The rationale behind utilizing weakly-augmented views for generating pseudo-labels stems from the expectation that the
> model's predictions on these mildly augmented versions will exhibit greater confidence compared to
> predictions from strongly augmented counterparts.
> The substantial alterations introduced by strong augmentations may distort the inherent data structure of the
> image, potentially introducing noise to the predictions.
> Conversely, when pseudo-labels are generated from weakly-augmented views, they tend to be less noisy,
> facilitating smoother and more effective guidance of the learning process.
>
> [1] FixMatch: Simplifying Semi-Supervised Learning with Consistency and Confidence, Sohn et al., NeurIPS 2020.

---

### Official Review · Reviewer_bBjN · 2023-07-07

**Soundness:** 4 excellent
**Presentation:** 3 good
**Contribution:** 3 good
**Rating:** 7
**Confidence:** 5

**Summary:**

This paper proposed a new approach to improve zero-shot vision recognition capability which leveraged the common embedding space for image and text. Specifically, with a pre-trained VLM (vision-language model), the authors leveraged LLM to automatically generate multiple language prompts for training a text-based classifier, which can be adapted for vision classification thanks to the shared embedding space. The classifier is further applied to augmented images to provide pseudo labels for finetuning the visual encoder in a label-free manner (pseudo labels generated from the text classifier). The authors perform extensive experiments to validate the effectiveness of the proposed training paradigm and on multiple benchmarks the model achieved significant performance gain.

**Strengths:**

1. The proposed method is intuitive and theoretically sound. The shared image-text embedding space naturally serves as a bridge for applying text-trained classifiers on image domains. The simple idea turns out to be very effective and brings significant performance boost across multiple datasets.
2. The conclusion is supported by extensive experiments with various ablation studies, showing many interesting patterns.
3. The paper is well-written and in good shape, it's easy to read.

**Weaknesses:**

1. Section 3.2 is named as "unsupervised finetuning" but technically since the model is trained with smoothed cross entropy loss, it's not truly "unsupervised", just that the labels are from text-based classifier (thus "pseudo labels"). The name is somewhat misleading as audience may get confused with unsupervised approaches.
2. CLIP is pre-trained with massive data thus demonstrating good zero-shot capability. Although this paper experimented on multiple benchmark datasets, the images of these datasets are all natural thus similar in style. The proposed approach achieves significant performance gain without real labels but the assumption is that the pseudo labels are relatively accurate due to image consistency. Therefore it is better to emphasize the point at least as a potential limitations. It might be better to be more careful with some claims e.g. L317 regarding the cross-modality transfer capability.

**Questions:**

1. Some other recent work used similar ideas for finetuning VLMs so maybe it's worth mentioning and provide a comparison. For example, [1] used a similar idea to generate multiple text and image prompts for finetuning VLMs domain adaptation.

[1] https://arxiv.org/abs/2306.16658

**Limitations:**

The paper has a dedicated section discussing about the limitations with the single linear layer of text classifier. Some more important limitations may need emphasizing as I suggested in the weaknesses section that the general assumption of the effectiveness of the proposed approach is the high quality of the underlying VLM model. No obvious potential negative societal impact.

---

> ### Author Rebuttal · Authors · 2023-08-08
>
> Thank you for the time and effort spent in reviewing our paper. In the following, we provide a response to the questions raised in the review.
>
> **Unsupervised finetuning naming convention.** Thank you for pointing it out. In our manuscript, we refer to the second
> stage of LaFTer (Section 3.2 of the originally submitted manuscript) as 'Unsupervised Finetuning on Target Domain Images'
> because we do not use any _ground truth_ labels from the training samples of the downstream datasets.
> The supervision in the Cross-Entropy Loss is generated by the output predictions of the model itself.
> However, we do acknowledge the source of confusion, which might be caused
> because 'Unsupervised Finetuning' has been used in a variety of contexts, more commonly for
> 'Unsupervised Representation Learning'. Following the suggestion, we will consider changing the name of Section 3.2 to
> 'Label-Free Finetuning on Target Domain Images'.
> We would also be happy to incorporate any further suggestions which may be provided during the discussion period.
>
> **Distribution of images of the downstream datasets.** In addition to experimenting with our LaFTer on datasets
> containing images collected in the real-world (e.g., CIFAR, ImageNet, SUN-397, UCF-101), we also tested our method
> on out-of-distribution variant of the original ImageNet dataset (ImageNet-Adversarial), which contains adversarial images for the original categories in the ImageNet dataset, as well as on satellite data (EuroSat), which has images from space that are
> likely very rare in CLIP web train data. Please note that our method shows considerable performance gains by adapting
> to the out-of-distribution images as well. For example, for EuroSat we obtain a performance gain of $28.8$% over the
> original CLIP model. However, we do agree that since CLIP is trained on a large corpus of data scraped from the
> internet, it is extremely challenging to know the _true_ data distribution of the training set and hence, it might
> be the case that CLIP has seen similar images to the out-of-distribution variants during its large-scale pre-training.
> We will emphasize this point and also rephrase $\text{L}317$ in the limitation section, of the revised manuscript.
>
> **Recent work for finetuning VLMs.** Thank you for highlighting this relevant work.
> Please note that it was released on ArXiv on $29^{th}$ of June, 2023 (long after the NeurIPS deadline).
> However, we will discuss (and compare, provided the code is open-sourced) this work in the related work section
> of our revised manuscript.

---

> > ### Comment · Reviewer_bBjN · 2023-08-16
> > **Thanks for the response!**
> >
> > I've read the response from authors and are mostly satisfied with the answers. I don't have other questions at this moment and would like to keep my rating for acceptance.

---

### Official Review · Reviewer_YPJ1 · 2023-07-07

**Soundness:** 3 good
**Presentation:** 3 good
**Contribution:** 2 fair
**Rating:** 5
**Confidence:** 4

**Summary:**

This paper proposes a new method to improve the zero-shot classification accuracy for a pre-trained vision-language (VL) model. By leveraging the shared embedding space between the vision and language modalities, and relatively accessible text data generated by large language models (LLMs), the proposed method trains a classifier on the embedding of generated texts for the zero-shot classes, which can also work on the image features. The proposed method also includes a pseudo-label strategy using the trained classifier to augment unlabeled image data. The paper shows compelling performance improvement over the original VL model and other non-label fine-tuning methods.

**Strengths:**

1. Overall I think it is an interesting idea. By using the shared embedding from the two modalities, it generates relatively "cheap" data by using an LLM to train a classifier directly for the target classes. This classifier further gives a better "starting point" for the pseudo-labeling in the next stage.
2. The evaluations on the benchmark datasets show considerable improvements over other label-free methods. It also outperforms 1-shot fine-tuning methods on certain datasets.

**Weaknesses:**

1. What bothers me the most is that the performance of only using Llama descriptions is quite underwhelming (Table 3). It has a big gap to only using class names. Even the pure GPT description is only 1% better than class names on average. I think the failure of Llama needs more explanations. Also, these results might imply: 1) the generated texts are not that effective; 2) you need a really good LLM (GPT3 or better) to generate decent text descriptions to make this method work. IMO, both points will limit the contribution and impact of the proposed method.
2. I feel the diversity and quality of the generated texts will be very important in the proposed approach, as it will affect the generalization ability of the trained classifier. It is better to have some ablations on these aspects.

**Questions:**

LLM can just generate random text descriptions of the class. For example, it might generate the history of an object which is not very useful in visual classification. Did you consider imposing any constraints to make the LLMs generate only visually meaningful texts?

**Limitations:**

To me, the biggest limitations are
1. The method might rely on a super good LLM which might not be accessible in many real-world scenarios.
2. The generated text for visual classification might not be visually meaningful.
No negative societal impact was observed.

---

> ### Author Rebuttal · Authors · 2023-08-08
>
> Thank you for the time and effort spent in reviewing our paper. In the following, we provide a response to the questions raised in the review.
>
> **Failure of Llama needs more explanation.** While comparing the descriptions of CIFAR-10 classes generated by Llama and GPT respectively, we found the following differences which could have an effect on the eventual classification performance.
>
> - In most cases, GPT mentions the class name in its descriptions, while Llama does not. Our experiments (in Table 1 of global response), highlight that this property influences the results.
>
> - The Llama descriptions often contain technical terms (e.g., in $\sim53$% of the reviewed descriptions), while GPT descriptions use more common terms (e.g., in $\sim94$% of the reviewed descriptions). To describe the airplane category, Llama mentioned terms like empennage, livery, composite material, whereas GPT descriptions contain terms, such as tires, metal plates etc. As (intuitively) technical terms were rarely encountered during the CLIP text encoder pre-training (e.g., the airplane class contained only $\sim8$% technical terms in the reviewed LAOIN-400m [1] captions, which can be a representative of CLIP alt-text), it might be harder for it to leverage those terms for classification.
>
> - The Llama descriptions had a larger variety of sentence structures than GPT, combining technical information with descriptive elements, whereas the GPT used simpler sentence structure with less variety, mentioning important details in a more succinct and structured manner. As the CLIP text encoder is trained on web collection of images and their alt-text, it is logical that it responds better to tuning with simpler sentence structure.
>
> To test the benefits of including the class names, we prepend the class names to each of the Llama descriptions and provide results in Table 1 of the global response.
> We see that by simply prepending the class name the performance increased by $3.8$% for CIFAR-100 and the average result is also $0.8$% better than the results obtained by using the GPT descriptions (while the results obtained by using descriptions from both the LLMs are on average $2.8$% better than using class names alone).
>
> **Generated descriptions might not be so effective.** In Table $3$ of the main manuscript, we see that the GPT descriptions perform well on datasets composed of naturally occurring images (CIFAR-10/100 and ImageNet), showing $4.0$% increase on CIFAR-100 and $2.1$% increase on the large-scale ImageNet, as compared to using only the class names.
>
> On the out-of-distribution datasets like ImageNet-Adversarial (A) and ImageNet-Rendition (R), GPT descriptions show a degradation.
> For ImageNet-A it is expected, as learning a visual classifier with detailed LLM descriptions of classes enhances attention to those class details, while ImageNet-A contains adversarial images collected such that details of wrong classes appear in the images to confuse the classifier. On ImageNet-R, composed of renditions of ImageNet classes, we can easily improve the performance by slightly changing the LLM prompts to generate class descriptions for text-only pre-training in our LaFTer. For example, we prompt the LLMs (Llama and GPT) to provide us with descriptions of different types of renditions of objects present in the ImageNet-R dataset (e.g., art, graffiti, embroidery). An example prompt to the LLM is:
>
> - Describe what an _embroidery rendition_ of a {category} looks like.
>
> By using these _targeted descriptions_ for the datasets, we gain substantial improvements. For GPT, we obtain an improvement of $4.9$% ($63.6$% $\to$ $68.5$%), and for Llama descriptions prepended with class names we gain $2.6$% improvement in accuracy ($63.5$% $\to$ $66.1$%). Generating dataset-specific descriptions opens interesting future work directions for cross-domain text-only tuning (as already indicated by our preliminary results on ImageNet-R). Since, the targeted responses help to mitigate distribution shifts from the text side, it is naturally reflected on the vision side due to the shared embedding space.
>
> **Method requires a really good LLM (GPT or better).** In our paper, we perform experiments with the open-source Llama and in the rebuttal discuss the reasons why the results obtained by using Llama descriptions lag behind GPT, and how they can be improved (Table 1 of global response).
> In an effort to improve the accessibility of LaFTer, we also perform experiments with recently open-sourced Llama-2 by META.
> The results (Table 1 of global response) show that the descriptions generated by Llama-2 provide competitive results to the GPT descriptions and even outperform them on average (by $1.2$%) when prepended with the class names. These results show that the recently open-sourced LLMs can be readily used as an alternative to GPT.
>
> **Diversity and quality of descriptions.** In Figure 1 of the global response, we plot the resulting accuracy after randomly sampling a certain amount of GPT descriptions per class. We observe that as we increase the amount of descriptions per class, the accuracy also increases, highlighting that the diversity of generated descriptions is indeed important. Furthermore, our experiments with generating _targeted descriptions_ for ImageNet-R (described above), demonstrate that the quality of responses from the LLM has a strong influence on the eventual classification performance.
>
> **Imposing constraints on LLMs for visually meaningful texts.**
> We expect that our prompts to the LLMs would impose conditions on the responses to produce visually meaningful texts. Two example prompts to the LLM are:
>
> - Describe what a {category} looks like?
> - How can you identify a {category}?
>
> These prompts instruct the LLM to produce _category descriptions_ in a visually meaningful way. Please refer to the Supplementary Section 3.1, for a complete list of prompts.
>
> [1] Schuhmann et al. LAION-400m, NeurIPS 2021.

---

> > ### Comment · Reviewer_YPJ1 · 2023-08-20
> >
> > Thank you for your response! Below are my comments:
> >
> > 1. The analysis of the generated texts from GPT and Llama seems interesting. Although I am not sure if you have a clear definition of "technical" and "common" terms when you calculate their frequency, I get the point. Because the downstream model is using CLIP, the generated texts need to be more similar to what it was trained with, otherwise, the model cannot understand it (e.g., a lot of "technical terms"). This makes sense and aligns with my hypothesis that the Llama generated texts were not that effective. These analyses are valuable to anyone who wants to use the proposed method so I recommend authors add them to the final revision. Also on this point, one thing you could try is to enforce/encourage the Llama to only generate texts using "common terms", by using a prompt (few-shot instruction) or a constrained vocabulary (if you have a clear definition of the "technical" terms).
> >
> > 2. Results with LLama2 and diversity ablation. I appreciate these results and they should be added to the final version. It is encouraging to see the method can work with more accessible open-sourced LLMs.
> >
> > Overall I am satisfied with the response as it addresses my concern about the accessibility of the capable LLMs for the proposed method. I feel comfortable increasing my score and encourage the authors to include the analysis and results in the response to the final version.

---

> > > ### Author Response · Authors · 2023-08-20
> > > **Thanks!**
> > >
> > > Thank you for the positive feedback on our response!
> > > We would certainly include the analysis and the results from the response in the revised version of the paper.
> > > Thanks again for your valuable suggestions!
> > > We would appreciate it if the reviewer could indeed increase the score (currently it seems to remain unchanged). Thanks!

---

### Author Rebuttal · Authors · 2023-08-08

We thank all the reviewers for their efforts to review our paper and for
providing insightful feedback.

We are happy to see that they found our work: **novel** `(YPJ1, zXQC, bBjN, o7C2)`,
**interesting** `(YPJ1, o7C2)` and **theoretically sound** `(bBjN)`.
Furthermore, we also thank them for highlighting that our work is **extensively evaluated**
on different benchmarks and shows **strong empirical results** `(YPJ1, bBjN, zXQC, o7C2)`,
contains **appropriate ablation studies** showing many interesting patterns and contributions of
individual components of the approach `(bBjN, zXQC)`, and also for finding our work
**easy to read and understand** `(bBjN, zXQC)`.

In the attached PDF, we present more insights to gain a further understanding of our LaFTer. We refer to this PDF throughout the response (to each reviewer)  as **`global response`**.

---

### Decision · Program_Chairs · 2023-09-21

**Decision:**

Accept (poster)

**Comment:**

The paper proposes a technique to train a classifier zero-shot (with no labelled data) using a pre-trained CLIP model and a language model to generate pseudo-labels. The reviews appreciate that the method was interesting and intuitive, as well as the breadth of the experiments and strength of the results. During rebuttal, the addition of adjustments to one of the baselines and discussion of the generated texts were useful. At the end of the discussion, the decision is a unanimous accept.